# STOCHASTIC INVERSE REINFORCEMENT LEARNING

## ABSTRACT

The goal of the inverse reinforcement learning (IRL) problem is to recover the reward functions from expert demonstrations. However, the IRL problem like any ill-posed inverse problem suffers the congenital defect that the policy may be optimal for many reward functions, and expert demonstrations may be optimal for many policies. In this work, we generalize the IRL problem to a well-posed expectation optimization problem *stochastic inverse reinforcement learning* (SIRL) to recover the probability distribution over reward functions. We adopt the Monte Carlo expectation-maximization (MCEM) method to estimate the parameter of the probability distribution as the first solution to the SIRL problem. The solution is *succinct, robust, and transferable* for a learning task and can generate alternative solutions to the IRL problem. Through our formulation, it is possible to observe the intrinsic property for the IRL problem from a *global* viewpoint, and our approach achieves a considerable performance on the *objectworld*.

## 1 INTRODUCTION

The IRL problem addresses an inverse problem that a set of expert demonstrations determines a reward function over a Markov decision process (MDP) if the model dynamics are known Russell (1998); Ng et al. (2000). The recovered reward function provides a *succinct, robust, and transferable* definition of the learning task and completely determines the optimal policy. However, the IRL problem is ill-posed that the policy may be optimal for many reward functions and expert demonstrations may be optimal for many policies. For example, all policies are optimal for a constant reward function. In a real-world scenario, experts always act sub-optimally or inconsistently, which is another challenge.

To overcome these limitations, two classes of probabilistic approaches for the IRL problem are proposed, i.e., Bayesian inverse reinforcement learning (BIRL) Ramachandran & Amir (2007) based on Bayesians' maximum a posteriori (MAP) estimation and maximum entropy IRL (MaxEnt) Ziebart et al. (2008); Ziebart (2010) based on frequentists' maximum likelihood (MLE) estimation. BIRL solves for the distribution of reward functions without an assumption that experts behave optimally, and encode the external *a priori* information in a choice of a prior distribution. However, BIRL also suffers from the practical limitation that a large number of algorithmic iterations is required for the procedure of Markov chain Monte Carlo (MCMC) in a sampling of posterior over reward functions. Advanced techniques, for example Kernel technique Michini & How (2012) and gradient method Choi & Kim (2011), are proposed to improve the efficiency and tractability of this situation.

MaxEnt employs the principle of maximum entropy to resolve the ambiguity in choosing demonstrations over a policy. This class of methods, inheriting the merits from previous non-probabilistic IRL approaches including Ng et al. (2000); Abbeel & Ng (2004); Ratliff et al. (2006); Abbeel et al. (2008); Syed & Schapire (2008); Ho et al. (2016), imposes regular structures of reward functions in a combination of hand-selected features. Formally, the reward function is a linear or nonlinear combination of the feature basis functions which consists of a set of real-valued functions $\{\phi_i(s, a)\}_{i=1}$ hand-selected by experts. The goal of this approach is to find the best-fitting weights of feature basis functions through the MLE approach. Wulfmeier et al. (2015) and Levine et al. (2011) use deep neural networks and Gaussian processes to fit the parameters based on demonstrations respectively but still suffer from the problem that the true reward shaped by the changing environment dynamics.

Influenced by the work of Finn et al. (2016a;b), Fu et al. (2017) propose a framework called adversarial IRL (AIRL) to recover robust reward functions in a changing dynamics based on adversarial learning and achieves superior results. Compared with AIRL, another adversarial method called generative

adversarial imitation learning (GAIL) Ho & Ermon (2016) seeks to directly recover the expert's policy rather than reward functions. Many follow-up methods enhance and extend GAIL for multipurpose in various application scenarios Li et al. (2017); Hausman et al. (2017); Wang et al. (2017). However, GAIL is in a lack of an explanation of expert's behavior and a portable representation for the knowledge transfer which are the merits of the class of the MaxEnt approach, because the MaxEnt approach is equipped with the "transferable" regular structures over reward functions.

In this paper, under the framework of the MaxEnt approach, we propose a generalized perspective of studying the IRL problem called *stochastic inverse reinforcement learning* (SIRL). It is formulated as an expectation optimization problem aiming to recover a probability distribution over the reward function from expert demonstrations. The solution of SIRL is *succinct* and *robust* for the learning task in the meaning that it can generate more than one weight over feature basis functions which compose alternative solutions to the IRL problem. Benefits of the class of the MaxEnt method, the solution to our generalized problem SIRL is also *transferable*. Since of the intractable integration in our formulation, we employ the Monte Carlo expectation-maximization (MCEM) approach Wei & Tanner (1990) to give the first solution to the SIRL problem in a model-based environment. In general, the solutions to the IRL problem are not always best-fitting in the previous approaches because a highly nonlinear inverse problem with the limited information is very likely to get trapped in a secondary maximum in the recovery. Taking advantage of the Monte Carlo mechanism of a *global* exhaustive search, our MCEM approach avoids the secondary maximum and theoretically convergent demonstrated by pieces of literature Caffo et al. (2005); Chan & Ledolter (1995). Our approach is also quickly convergent because of the preset simple geometric configuration over weight space in which we approximate it with a Gaussian Mixture Model (GMM). Hence, our approach works well in a real-world scenario with a small and variability set of expert demonstrations.

In particular, the contributions of this paper are threefold:

1. We generalize the IRL problem to a well-posed expectation optimization problem SIRL.

2. We provide the first theoretically existing solution to SIRL by the MCEM approach.

3. We show the effectiveness of our approach by comparing the performance of the proposed method to those of the previous algorithms on the *objectworld*.

## 2 PRELIMINARY

An MDP is a tuple $\mathcal{M} := \langle \mathcal{S}, \mathcal{A}, \mathcal{T}, R, \gamma \rangle$, where $\mathcal{S}$ is the set of states, $\mathcal{A}$ is the set of actions, and the transition function (a.k.a. model dynamics) $\mathcal{T} := \mathbb{P}(s_{t+1} = s' | s_t = s, a_t = a)$ for $s, s' \in \mathcal{S}$ and $a \in \mathcal{A}$ is the probability of being current state $s$, taking action $a$ and yielding next state $s'$. Reward function $\mathcal{R}(s, a)$ is a real-valued function and $\gamma \in [0, 1)$ is the discount factor. A policy $\pi : \mathcal{S} \to \mathcal{A}$ is deterministic or stochastic, where the deterministic one is written as $a = \pi(s)$, and the stochastic one is as a conditional distribution $\pi(a|s)$. Sequential decisions are recorded in a series of episodes which consist of states $s$, actions $a$, and rewards $r$. The goal of reinforcement learning aims to get optimal policy $\pi^*$ for maximizing the expected total reward, i.e.

$$\pi^* := \arg\max_{\pi} \mathbb{E}\Big[ \sum_{t=0}^{\infty} \gamma^t \cdot \mathcal{R}(s_t, a_t) \Big| \pi \Big].$$

Given an MDP without a reward function $\mathcal{R}$, i.e., MDP\R=$\langle \mathcal{S}, \mathcal{A}, \mathcal{T} \rangle$, and $m$ expert demonstrations $\zeta^E := \{\zeta^1, \cdots, \zeta^m\}$. Each expert demonstration $\zeta^i$ is a sequential of state-action pairs. The goal of the IRL problem is to estimate the unknown reward function $\mathcal{R}(s, a)$ from expert demonstrations $\zeta^E$ and the reward functions compose a complete MDP. The estimated complete MDP yields an optimal policy that acts as closely as the expert demonstrations.

### 2.1 REGULAR STRUCTURE OF REWARD FUNCTIONS

In this section, we provide a formal definition of the regular (linear/nonlinear) structure of reward functions. The linear structure Ng et al. (2000); Ziebart et al. (2008); Syed & Schapire (2008); Ho

et al. (2016) is an $M$ linear combination of feature basis functions, written as,

$$\mathcal{R}(s, a) := \sum_{i}^{M} \alpha_i \cdot \phi_i(s, a),$$

where $\phi_i : \mathcal{S} \times \mathcal{A} \longmapsto \mathbb{R}^d$ are a $d$-dimensional feature functions hand-selected by experts.

The nonlinear structure Wulfmeier et al. (2015) is of the form as follows,

$$\mathcal{R}(s, a) := \mathcal{N}\big(\phi_1(s, a), \cdots, \phi_M(s, a)\big),$$

where $\mathcal{N}$ is neural networks of hand-crafted reward feature basis functions $\{\phi_i(s, a)\}_{i=1}^{M}$.

In the following section, we propose an optimization problem whose solution is a probability distribution over weights of basis functions $\{\phi_i(s, a)\}_{i=1}$.

## 2.2 PROBLEM STATEMENT

Given an MDP$\backslash R := (\mathcal{S}, \mathcal{A}, \mathcal{T}, \gamma)$ with a known transition function $\mathcal{T} := \mathbb{P}(s_{t+1} = s' | s_t = s, a_t = a)$ for $s, s' \in \mathcal{S}$ and $a \in \mathcal{A}$ and a hand-crafted reward feature basis function $\{\phi_i(s)\}_{i=1}^{M}$. A stochastic regular structure on the reward function assumes weights $\mathcal{W}$ of the reward feature functions $\phi_i(s)$, which are random variables with a reward conditional probability distribution $\mathcal{D}^{\mathcal{R}}(\mathcal{W}|\zeta^E)$ conditional on expert demonstrations $\zeta^E$. Parametrizing $\mathcal{D}^{\mathcal{R}}(\mathcal{W}|\zeta^E)$ with parameter $\Theta$, our aim is to estimate the best-fitting parameter $\Theta^*$ from the expert demonstrations $\zeta^E$, such that $\mathcal{D}^{\mathcal{R}}(\mathcal{W}|\zeta^E, \Theta^*)$ more likely generates weights to compose reward functions as the ones derived from expert demonstrations, which is called *stochastic inverse reinforcement learning* problem.

Suppose a representative trajectory class $\mathcal{C}_\epsilon^E$ satisfies that each trajectory element set $\mathcal{O} \in \mathcal{C}_\epsilon^E$ is a subset of expert demonstrations $\zeta^E$ with the cardinality at least $\epsilon \cdot |\zeta^E|$, where $\epsilon$ is a preset threshold and $|\zeta^E|$ is the number of expert demonstrations, written as,

$$\mathcal{C}_\epsilon^E := \big\{\mathcal{O}\big|\mathcal{O} \subset \zeta^E \text{ with } |\mathcal{O}| \geq \epsilon \cdot |\zeta^E|\big\}.$$

Integrate out unobserved weights $\mathcal{W}$, and the SIRL problem is formulated to estimate parameter $\Theta$ on an expectation optimization problem over the representative trajectory class as follows:

$$\Theta^* := \arg\max_{\Theta} \mathbb{E}_{\mathcal{O} \in \mathcal{C}_\epsilon^E} \Big[\int_{\mathcal{W}} f_{\mathcal{M}}(\mathcal{O}, \mathcal{W}|\Theta) d\mathcal{W}\Big], \tag{1}$$

where trajectory element set $\mathcal{O}$ assumes to be uniformly distributed for the sake of simplicity in this study, and $f_{\mathcal{M}}$ is the conditional joint probability density function of trajectory element $\mathcal{O}$ and weights $\mathcal{W}$ for reward feature functions conditional on parameter $\Theta$.

### 2.2.1 NOTE:

- Problem 1 is well-posed and typically not intractable analytically.
- Trajectory element set $\mathcal{O}$ is usually known from the rough estimation of the statistics in expert demonstrations in practice.
- We introduce *representative trajectory class* to overcome the limitation of the sub-optimality of expert demonstrations $\zeta^E$ Ramachandran & Amir (2007), which is also called a lack of sampling representativeness in statistics Kruskal & Mosteller (1979), e.g., driver's demonstrations encode his own preferences in driving style but may not reflect the true rewards of an environment. With the technique of representative trajectory class, each subset of driver's demonstrations (i.e. trajectory element set $\mathcal{O}$) constitutes a subproblem in Problem 1.

## 3 METHODOLOGY

In this section, we propose a novel approach to estimate the best-fitting parameter $\Theta^*$ in Problem 1, which is called the two-stage hierarchical method, *a variant of MCEM* method.

### 3.1 TWO-STAGE HIERARCHICAL METHOD

The two-stage hierarchical method requires us to write parameter $\Theta$ in a profile form $\Theta := (\Theta_1, \Theta_2)$. The conditional joint density $f_{\mathcal{M}}(\mathcal{O}, \mathcal{W}|\Theta)$ in Equation 1 can be written as the product of two conditional densities $g_{\mathcal{M}}$ and $h_{\mathcal{M}}$ as follows:

$$f_{\mathcal{M}}(\mathcal{O}, \mathcal{W}|\Theta_1, \Theta_2) = g_{\mathcal{M}}(\mathcal{O}|\mathcal{W}, \Theta_1) \cdot h_{\mathcal{M}}(\mathcal{W}|\Theta_2). \tag{2}$$

Take the log of both sides in Equation 2, and we have

$$\log f_{\mathcal{M}}(\mathcal{O}, \mathcal{W}|\Theta_1, \Theta_2) = \log g_{\mathcal{M}}(\mathcal{O}|\mathcal{W}, \Theta_1) + \log h_{\mathcal{M}}(\mathcal{W}|\Theta_2). \tag{3}$$

We maximize the right side of Equation 3 over the profile parameter $\Theta$ in the expectation-maximization (EM) update steps at the $t$-th iteration independently as follows,

$$\Theta_1^{t+1} := \arg\max_{\Theta_1} \mathbb{E}\Big( \log g_{\mathcal{M}}(\mathcal{O}|\mathcal{W}, \Theta_1)\Big|\mathcal{C}_\epsilon^E, \Theta^t \Big); \tag{4}$$

$$\Theta_2^{t+1} := \arg\max_{\Theta_2} \mathbb{E}\Big( \log h_{\mathcal{M}}(\mathcal{W}|\Theta_2)\Big|\Theta^t \Big). \tag{5}$$

#### 3.1.1 INITIALIZATION

We randomly initialize profile parameter $\Theta^0 := (\Theta_1^0, \Theta_2^0)$ and sample a collection of $N_0$ rewards weights $\{\mathcal{W}_1^{\Theta^0}, \cdots, \mathcal{W}_N^{\Theta^0}\} \sim \mathcal{D}^{\mathcal{R}}(\mathcal{W}|\zeta^E, \Theta_2^0)$. The reward weights $\mathcal{W}_i^{\Theta^0}$ compose reward $R_{\mathcal{W}_i^{\Theta^0}}$ in each learning task $\mathcal{M}_i^0 := (\mathcal{S}, \mathcal{A}, \mathcal{T}, R_{\mathcal{W}_i^{\Theta^0}}, \gamma)$ for $i = 1, \cdots, N_0$.

#### 3.1.2 FIRST STAGE

In the first stage, we aim to update parameter $\Theta_1$ in the intractable expectation of Equation 4. Specifically, we take a Monte Carlo method to estimate model parameters $\Theta_1^{t+1}$ in an empirical expectation at the $t$-th iteration,

$$\mathcal{E}\big[ \log g_{\mathcal{M}}(\mathcal{O}|\mathcal{W}, \Theta_1^{t+1})\big|\mathcal{C}_\epsilon^E, \Theta^t \big] := \frac{1}{N_t} \cdot \sum_{i=1}^{N_t} \log g_{\mathcal{M}_i^t}(\mathcal{O}_i^t|\mathcal{W}_i^{\Theta^t}, \Theta_1^{t+1}), \tag{6}$$

where reward weights at the $t$-th iteration $\mathcal{W}_i^{\Theta^t}$ are randomly drawn from the reward conditional probability distribution $\mathcal{D}^{\mathcal{R}}(\mathcal{W}|\zeta^E, \Theta^t)$ and compose a set of learning tasks $\mathcal{M}_i^t := (\mathcal{S}, \mathcal{A}, \mathcal{T}, R_{\mathcal{W}_i^{\Theta^t}}, \gamma)$ with a trajectory element set $\mathcal{O}_i^t$ uniformly drawn from representative trajectory class $\mathcal{C}_\epsilon^E$, for $i = 1, \cdots, N_t$.

The parameter $\Theta_1^{t+1}$ in Equation 6 has $N_t$ coordinates written as $\Theta_1^{t+1} := \Big((\Theta_1^{t+1})_1, \cdots, (\Theta_1^{t+1})_{N_t}\Big)$. For each learning task $\mathcal{M}_i^t$, the $i$-th coordinate $(\Theta_1^{t+1})_i$ is derived from maximization of a posteriori,

$$(\Theta_1^{t+1})_i := \arg\max_{\theta} \log g_{\mathcal{M}_i^t}\big(\mathcal{O}_i^t|\mathcal{W}_i^{\Theta^t}, \theta\big),$$

which is a convex formulation maximized by a gradient ascent method.

In practice, we move $m$ steps uphill to the optimum in each learning task $\mathcal{M}_i^t$. The update formula of $m$-step reward weights $\mathcal{W}^m{}_i^{\Theta^t}$ is written as

$$\mathcal{W}^m{}_i^{\Theta^t} := \mathcal{W}_i^{\Theta^t} + \sum_{i=1}^{m} \lambda_i^t \cdot \nabla_{(\Theta_1)_i} \log g_{\mathcal{M}_i^t}\big(\mathcal{O}_i^t|\mathcal{W}_i^{\Theta^t}, (\Theta_1)_i\big),$$

where the learning rate $\lambda_i^t$ at the $t$-th iteration is preset. Hence, the parameter $\Theta_1^{t+1}$ is represented as $\Theta_1^{t+1} := \Big(\mathcal{W}^m{}_1^{\Theta^t}, \cdots, \mathcal{W}^m{}_{N_t}^{\Theta^t}\Big)$.

### 3.1.3 SECOND STAGE

In the second stage, we aim to update parameter $\Theta_2$ in the intractable expectation of Equation 5. Specifically, we consider the empirical expectation at the $t$-th iteration as follows,

$$\mathcal{E}\big(\log h_{\mathcal{M}}(\mathcal{W}|\Theta_2^{t+1})\big|\Theta^t\big) := \frac{1}{N_t} \cdot \sum_{i=1}^{N_t} \log h_{\mathcal{M}_i^t}(\mathcal{W}^{m\Theta^t}_i|\Theta_2^{t+1}), \qquad (7)$$

where $h_{\mathcal{M}}$ is implicit but fitting a set of $m$-step reward weights $\{\mathcal{W}^{m\Theta^t}_i\}_{i=1}^{N_t}$ in a generative model yields a large empirical expectation value. The reward conditional probability distribution $\mathcal{D}^{\mathcal{R}}(\mathcal{W}|\zeta^E, \Theta_2^{t+1})$ is a generative model formulated as a Gaussian Mixture Model (GMM), i.e.

$$\mathcal{D}^{\mathcal{R}}(\mathcal{W}|\zeta^E, \Theta_2^{t+1}) := \sum_{k=1}^{K} \alpha_k \cdot \mathcal{N}(\mathcal{W}|\mu_k, \Sigma_k),$$

where $\alpha_k \geq 0$ and $\sum_{k=1}^{K} \alpha_k = 1$, and parameter set $\Theta_2^{t+1} := \{\alpha_k; \mu_k, \Sigma_k\}_{k=1}^{K}$.

We estimate parameter $\Theta_2^{t+1}$ in GMM by EM approach and initialize GMM with the $t$-th iteration parameter $\Theta_2^t$ with the procedure as follows:

For $i = 1, \cdots, N_t$, we have

- Expectation Step: Compute responsibility $\gamma_{ij}$ for $m$-step reward weight $\mathcal{W}^{m\Theta^t}_i$,

$$\gamma_{ij} := \frac{\alpha_j \cdot \mathcal{N}(\mathcal{W}^{m\Theta^t}_i|\mu_j, \Sigma_j)}{\sum_{k=1}^{K} \alpha_k \cdot \mathcal{N}(\mathcal{W}^{m\Theta^t}_i|\mu_k, \Sigma_k)}.$$

- Maximization Step: Compute weighted mean $\mu_j$ and variance $\Sigma_j$ by,

$$\mu_j := \frac{\sum_{i=1}^{N_t} \gamma_{ij} \cdot \mathcal{W}^{m\Theta^t}_i}{\sum_{i=1}^{N_t} \gamma_{ij}}; \quad \alpha_j := \frac{1}{N_t} \cdot \sum_{i=1}^{N_t} \gamma_{ij}; \quad \Sigma_j := \frac{\sum_{i=1}^{N_t} \gamma_{ij} \cdot (\mathcal{W}^{m\Theta^t}_i - \mu_j) \cdot (\mathcal{W}^{m\Theta^t}_i - \mu_j)^T}{\sum_{i=1}^{N_t} \gamma_{ij}}.$$

After EM converges, $\Theta_2^{t+1} := \{\alpha_k; \mu_k, \Sigma_k\}_{k=1}^{K}$ and profile parameter $\Theta^{t+1} := (\Theta_1^{t+1}, \Theta_2^{t+1})$.

Finally, when the two-stage hierarchical method converges, parameter $\Theta_2$ of profile parameter $\Theta$ is our desired best-fitting parameter $\Theta^*$ for $\mathcal{D}^{\mathcal{R}}(\mathcal{W}|\zeta^E, \Theta^*)$.

### 3.2 TERMINATION CRITERIA

In this section, we discuss the termination criteria in our algorithm. EM terminates usually when the parameters do not substantively change after enough iterations. For example, one classic termination criterion in EM terminates at the $t$-th iteration satisfying as follows,

$$\max \frac{|\theta^t - \theta^{t-1}|}{|\theta^t| + \delta_{EM}} < \epsilon_{EM},$$

for user-specified $\delta_{EM}$ and $\epsilon_{EM}$, where $\theta$ is the model parameter in EM.

However, the same termination criterion for MCEM has a risk of early terminating because of the Monte Carlo error in the update step. Hence, we adopt a practical method in which the following stopping criterion holds in three consecutive times,

$$\max \frac{|\Theta^t - \Theta^{t-1}|}{|\Theta^t| + \delta_{MCEM}} < \epsilon_{MCEM},$$

for user-specified $\delta_{MCEM}$ and $\epsilon_{MCEM}$ Booth & Hobert (1999). Other stopping criteria for MCEM refers to Caffo et al. (2005); Chan & Ledolter (1995).

### 3.3 CONVERGENCE ISSUE

The convergence issue of MCEM is more complicated than EM. In light of model-based interactive MDP\R, we can always increase the sample size during each iteration of MCEM. In practice, we require the Monte Carlo sample size satisfy the following inequality,

$$\sum_t \frac{1}{N_t} < \infty.$$

Additional requirement for the convergence property is a compact assumption over that parameter space. A comprehensive proof refers to Chan & Ledolter (1995); Fort et al. (2003).

The pseudocode is given in Appendix.

## 4 EXPERIMENTS

We evaluate our approach on a classic environment *objectworld* introduced by Levine et al. (2011) which is a particularly challenging environment with a large number of irrelevant features and the highly nonlinearity of the reward functions. Note that since almost only *objectworld* provides a tool that allows analysis and display the evolution procedure of the SIRL problem in a 2D heat map, we skip the typical invisible physics-based control tasks for the evaluation of our approach, i.e. cartpole Barto et al. (1983), mountain car Moore (1990), MuJoCo Todorov et al. (2012), and etc.

We employ the expected value difference (EVD) proposed by Levine et al. (2011) to be the metric of optimality as follows:

$$\text{EVD}(\mathcal{W}) := \mathbb{E}\big[\sum_{t=0}^{\infty} \gamma^t \cdot R(s_t, a_t)\big|\pi^*\big] - \mathbb{E}\big[\sum_{t=0}^{\infty} \gamma^t \cdot R(s_t, a_t)\big|\pi(\mathcal{W})\big],$$

which is a measure of the difference between the expected reward earned under the optimal policy $\pi^*$, given by the true reward, and the policy derived from the rewards sampling from our reward conditional probability distribution $\mathcal{D}(\mathcal{W}|\zeta^E, \Theta^*)$, where $\Theta^*$ is the best estimation parameter in our approach.

### 4.1 OBJECTWORLD

The *objectworld* is a learning environment for the IRL problem. It is an $N \times N$ grid board with colored objects placed in randomly selected cells. Each colored object is assigned one inner color and one outer color from $C$ preselected colors. Each cell on the grid board is a state, and stepping to four neighbor cells (up, down, left, right) or staying in place (stay) are five actions with a 30% chance of moving in a random direction.

The ground truth of the reward function is defined in the following way. Suppose two primary colors of $C$ preselected colors are red and blue. The reward of a state is 1 if the state is within 3 steps of an outer red object and 2 steps of an outer blue object, -1 if the state is within 3 steps of an outer red object, and 0 otherwise. The other pairs of inner and outer colors are distractors. Continuous and discrete versions of feature basis functions are provided. For the continuous version, $\phi(s)$ is a $2C$-dimensional real-valued feature vector. Each dimension records the Euclidean distance from the state to objects. For example, the first and second coordinates are the distances to the nearest inner and outer red object respectively, and so on through all $C$ colors. For the discrete version, $\phi(s)$ is a $(2C \cdot N)$-dimensional binary feature vector. Each $N$-dimensional vector records a binary representation of the distance to the nearest inner or outer color object with the $d$-th coordinate 1 if the corresponding continuous distance is less than $d$.

### 4.2 EVALUATION PROCEDURE AND ANALYSIS

In this section, we design three tasks to evaluate the effectiveness of our generative model *reward conditional probability distribution* $\mathcal{D}(\mathcal{W}|\zeta^E, \Theta^*)$. For each task, the environment setting is as follows. The instance of a $10 \times 10$ *objectworld* has 25 random objects with 2 colors and a 0.9 discount factor. 20 expert demonstrations are generated according to the given optimal policy for the

recovery. The length of each expert demonstration is 5-grid size trajectory length. Four algorithms for the evaluation includes MaxEnt, DeepMaxEnt, SIRL, and DSIRL, where SIRL and DSIRL are implemented as Algorithm 2 in Appendix. The weights are drawn from reward conditional probability distribution $\mathcal{D}(\mathcal{W}|\zeta^E, \Theta^*)$ as the coefficients of feature basis functions $\{\phi_i(s,a)\}_{i=1}$.

In our evaluation, SIRL and DSIRL start from 10 samples and double the sample size per iteration until it converges for the convergence issue, refer to Section 3.2. In the first stage, the epochs of algorithm iteration are set to 20 and the learning rates are 0.01. The parameter $\epsilon$ in representative trajectory set $\mathcal{O}_\epsilon^E$ is preset as 0.95. In the second stage, 3-component GMM for SIRL and DSIRL is set with at most 1000 iterations before convergence. Additionally, the architecture of neural networks in DeepMaxEnt and DSIRL are implemented as 3-layer fully-connected with the sigmoid function.

### 4.2.1 EVALUATION PLATFORM

All the methods are implemented in Python 3.5 and Theano 1.0.0 with a machine learning distributed framework *Ray* Moritz et al. (2018). The experiments are conducted on a machine with Intel(R) Core(TM) i7-7700 CPU @ 3.60GHz and Nvidia GeForce GTX 1070 GPU.

### 4.2.2 RECOVERY EXPERIMENT

In the recovery experiment, we compare the true reward function, the optimal policy, and the optimal value with ones derived from expert demonstrations under the four methods. Since our approach is an average of all the outcomes which are prone to imitate the true optimal value from expert demonstrations, we use the mean of reward conditional probability distribution for SIRL and DSIRL as a comparison.

In Figure 1, the EVD of the optimal values in the last row are 48.9, 31.1, 33.7 and 11.3 for four methods respectively, and the covariances of GMM model for SIRL and DSIRL are limited up to 5.53 and 1.36 on each coordinate respectively. It yields that in a highly nonlinear inverse problem, the recovery abilities of SIRL and DISRL are better than MaxEnt's and DeepMaxEnt's respectively. The reason is mainly because Monte Carlo mechanism in our approach alleviates the problem of getting stuck in local minima by allowing random exit from it.

### 4.2.3 ROBUSTNESS EXPERIMENT

In the robustness experiment, we evaluate the robustness of our approach that solutions generated by $\mathcal{D}(\mathcal{W}|\zeta^E, \Theta^*)$ are effective to the IRL problem. To capture the robust solutions, we design the following generative algorithm with the pseudocode in Algorithm 1.

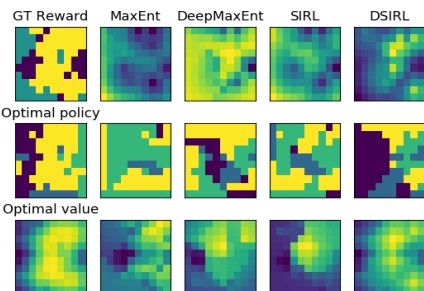
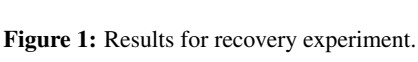

**Figure 1:** Results for recovery experiment.

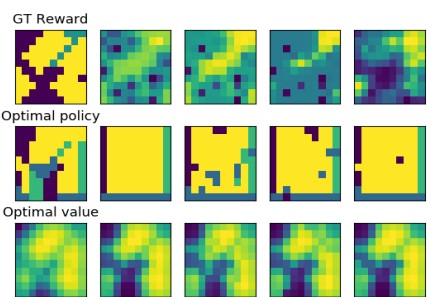

**Figure 2:** Results for the robustness experiment.

In the right generative algorithm, we use Frobenius norm to measure the distance between weights drawn from $\mathcal{D}(\mathcal{W}|\zeta^E, \Theta^*)$ as follows,

$$||\mathcal{W}||_{\mathcal{F}} := \sqrt{\mathrm{Tr}(\mathcal{W} \cdot \mathcal{W}^T)}.$$

We also constrain that each drawn weight $\mathcal{W} \sim \mathcal{D}(\mathcal{W}|\zeta^E, \Theta^*)$ in the solution set $\mathcal{G}$ satisfies as follows,

$$||\mathcal{W} - \mathcal{W}'||_{\mathcal{F}} > \delta \text{ and } \mathrm{EVD}(\mathcal{W}) < \epsilon,$$

where $\mathcal{W}'$ represents any member in the solution set $\mathcal{G}$. $\delta, \epsilon$ are the preset thresholds in the generative algorithm.

---

**Algorithm 1:** Generative Algorithm

**Input:** $\mathcal{D}(\mathcal{W}|\zeta^E, \Theta^*)$, required solution set size $N$, and preset thresholds $\epsilon$ and $\delta$.
**Output:** Solution set $\mathcal{G} := \{\mathcal{W}_i\}_{i=1}^N$.

---

**while** $i < N$ **do**
    $\mathcal{W} \sim \mathcal{D}(\mathcal{W}|\zeta^E, \Theta^*)$ ;
    **for** *any* $\mathcal{W}' \in \mathcal{S}$ **do**
        **if** $||\mathcal{W} - \mathcal{W}'||_{\mathcal{F}} > \delta$ *and* $EVD(\mathcal{W}) < \epsilon$ **then**
            $\mathcal{G} \leftarrow \mathcal{W}$;
        **end**
    **end**
    $i \leftarrow i + 1$;
**end**

---

In Figure 2, the right column figures are generated from weights in the solution set $\mathcal{G}$ whose EVD values are around 10.2. Note that the recovered reward function in the first row has a similar but different pattern appearance. The optimal value derived from these recovered reward functions has a very small EVD value with the true reward. It yields the effectiveness of our *robust* generative model which can generate more than one solutions to the IRL problem.

### 4.2.4 HYPERPARAMETER EXPERIMENT

In the hyperparameter experiment, we evaluate the effectiveness of our approach under the influence of different preset quantities and qualities of expert demonstrations. The amount of information carried in expert demonstrations composes a specific learning environment, and hence it has an impact on the effectiveness of our generative model. We verify three hyperparameters including the number of expert demonstrations in Figure 3, the trajectory length of expert demonstrations in Figure 4 and the portion size in representative trajectory class $\mathcal{C}_\epsilon^E$ in Figure 5 on the *objectworld*. The shadow of the line in the figures represents the standard error for each experimental trail. Notice that the EVDs for SIRL and DSIRL are both decreasing as the number and the trajectory length of expert demonstrations, and the portion size in the representative trajectory class are increasing. A notable point in Figure 3 is that very few expert demonstrations (less than 200) for our approach also yields a small EVDs, which manifests the merit of Monte Carlo mechanism in our approach.

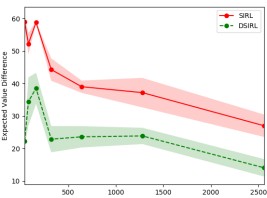
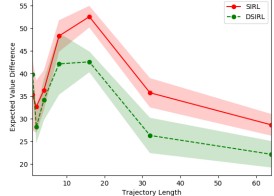
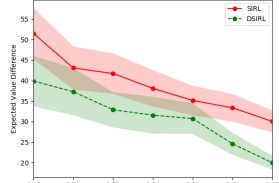

**Figure 3:** Results under 40, 80, 160, 320, 640, 1280 and 2560 expert demonstrations.

**Figure 4:** Results under 1, 2, 4, 8, 16, 32, and 64 grid size trajectory length of expert demonstrations.

**Figure 5:** Results under 0.65, 0.70, 0.75, 0.80, 0.85, 0.90, and 0.95 portion size in $\mathcal{C}_\epsilon^E$.

## 5 CONCLUSION

In this paper, we propose a generalized problem SIRL for the IRL problem to get the distribution of reward functions. The new problem is well-posed and we employ the method of MCEM to give the first *succinct, robust, and transferable* solution. In the experiment, we evaluate our approach on the *objectworld* and the experimental results confirm the effectiveness of our approach.

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

# A APPENDIX

The pseudocode of SIRL in Section 3 is as follows.

---

**Algorithm 2:** Stochastic Inverse Reinforcement Learning

---

**Input:** Model-based environment $(\mathcal{S}, \mathcal{A}, \mathcal{T})$ and expert demonstrations $\zeta^E$, Monte Carlo sample size $N_0$, and preset thresholds $\delta_{MCEM}$ and $\epsilon_{MCEM}$.
**Output:** Reward conditional probability distribution $\mathcal{D}^{\mathcal{R}}(\mathcal{W}|\zeta^E, \Theta^*)$.

---

**Initialization:** Randomly initialization of profile parameter $\Theta^0 := (\Theta_1^0, \Theta_2^0)$;
**while** *stopping criteria not satisfied (refer to Section 3.2)* **do**

    Draw $N_t$ reward weights $\mathcal{W}_i^{\Theta^t} \sim \mathcal{D}^{\mathcal{R}}(\mathcal{W}|\zeta^E, \Theta_2^t)$ to compose learning task $\mathcal{M}_i^t$ with uniformly drawn trajectory element set $\mathcal{O}_i^t$;
    # *First Stage: Monte Carlo estimation of weights for reward function*;
    **for** $\mathcal{M}_i^t$ **do**
        Evaluate $\nabla_{(\Theta_1)_i} \log g_{\mathcal{M}_i^t}(\mathcal{O}_i^t | \mathcal{W}_i^{\Theta^t}, (\Theta_1)_i)$ ;
        Compute updated weight parameter
        $\mathcal{W}^{m\Theta^t}_i \leftarrow \mathcal{W}_i^{\Theta^t} + \sum_{i=1}^m \lambda_i^t \cdot \nabla_{(\Theta_1)_i} \log g_{\mathcal{M}_i^t}(\mathcal{O}_i^t | \mathcal{W}_i^{\Theta^t}, (\Theta_1)_i)$ ;
    **end**
    Update $\Theta_1^{t+1} \leftarrow \{\mathcal{W}^{m\Theta^t}_i\}_{i=1}^{N_t}$ ;
    # *Second Stage: Fit GMM with $m$-step reward weight $\{\mathcal{W}^{m\Theta^t}_i\}_{i=1}^{N_t}$ with EM parameter initialization $\Theta_2^t$* ;
    **while** *EM not converge* **do**

        Expectation Step: $\gamma_{ij} \leftarrow \frac{\alpha_j \cdot \mathcal{N}(\mathcal{W}^{m\Theta^t}_i | \mu_j, \Sigma_j)}{\sum_{k=1}^K \alpha_k \cdot \mathcal{N}(\mathcal{W}^{m\Theta^t}_i | \mu_k, \Sigma_k)}$ ;
        Maximization Step:

$$\mu_j \leftarrow \frac{\sum_{i=1}^{N_t} \gamma_{ij} \cdot \mathcal{W}^{m\Theta^t}_i}{\sum_{i=1}^{N_t} \gamma_{ij}}; \quad \alpha_j \leftarrow \frac{1}{N_t} \cdot \sum_{i=1}^{N_t} \gamma_{ij};$$

$$\Sigma_j \leftarrow \frac{\sum_{i=1}^{N_t} \gamma_{ij} \cdot (\mathcal{W}^{m\Theta^t}_i - \mu_j) \cdot (\mathcal{W}^{m\Theta^t}_i - \mu_j)^T}{\sum_{i=1}^{N_t} \gamma_{ij}};$$

    **end**
    Update $\Theta_2^{t+1}$ and profile parameter $\Theta^{t+1} \leftarrow (\Theta_1^{t+1}, \Theta_2^{t+1})$;
**end**

---

