# OpenReview forum: "Stochastic Inverse Reinforcement Learning "
_ICLR.cc/2021/Conference — Reject_

### Official Review · AnonReviewer1 · 2020-10-16
**Confusing**

**Rating:** 2
**Confidence:** 5

**Review:**

Summary
--------
The paper describes a method for inverse reinforcement learning---called stochastic IRL---that learns a distribution over reward functions. In that sense, the method is similar to Bayesian approaches, however, the learned distribution doesn't seem to approximate the posterior for any given prior. It is hard to summarize the algorithm because it is highly unclear what the method actually does, but I will try my best by stating my most likely hypothesis.
The algorithm starts with an initial set of $N_0$ parameter vectors, each corresponding to the weights for a linear reward function. Each of these weights are improved by using a fixed number of gradient steps using MaxEnt-IRL (Ziebart et al. 2010), where different (overlapping) subsets of the demonstrations are used for the different weights. Subsequently, a GMM over weights is fitted to the $N_0$ improved weights using maximum likelihood (EM). This procedure is iterated, where the weights at each iteration are drawn from the current GMM. The number of weight vectors is doubled at every iteration t, i.e., $N_{t} = 2 N_{t-1}$.
I'm not at all confident that I got this right. For example, I solely inferred the use of MaxEnt-IRL from a sentence in the introduction ("In this paper, under the framework of the MaxEnt approach, [...]").
The algorithm is evaluated on a 10x10 gridworld and compared with MaxEnt-IRL and DeepMaxEnt-IRL. The experiments compare two versions of the algorithm called SIRL and DSIRL. The paper doesn't explain how they differ, but a neural network parameterization is mentioned for DSIRL, so I guess D stands for deep and uses DeepMaxEnt-IRL instead of MaxEnt-IRL.

Strong points
-------------
Learning a distribution over reward functions is a laudable goal for inverse reinforcement learning as it can be a principled way to deal with the uncertainty in modeling the expert.


Concerns
---------
__Clarity__
 The paper is really hard to follow, even though I'm very familiar with related work, and the algorithm and the equations seem to be quite simple. There are several reasons for this lack of clarity.
1) Lack of details. Actually not only details, some of the most fundamental aspects of the algorithm are not mentioned anywhere. For example, the likelihood of the demonstrations given the weights, $g(\mathcal{O}|\mathcal{W},\Omega)$ is nowhere defined. $\theta$ is also not defined.
2) Some of the equations are very confusing. For example, the second to last equations at page 4 seems to overload the subscript "i" which seems to index both the sampled weight as well as the gradient step, depending on the variable. Even when disentangling these different meanings the equation looks weird to me. Based on the surrounding text, I guess it should mean m gradient steps on the likelihood are performed, but the equation says different. It is also strange that the weights W are updated based on the gradient w.r.t. $\Omega_1$ and that $\Omega_1$ is set to the updated weights in the last equation at page 4. It seems like Omega_1 is superfluous. Also, the mixture model $\mathcal{D}(\mathcal{W}|\zeta, \Omega_2)$ seems to be conditional independent of $\zeta$ given $\Omega_2$. What's the difference between $\mathcal{D}(\mathcal{W}|\zeta, \Omega_2)$ and $h(\mathcal{W}|\Omega_2)$?
3) Some algorithmic choices are not motivated at all (for example, doubling the samples at every iteration), others are only (insufficiently) motivated in hindsight. For example, the use of "representative trajectory classes" is motivated after introducing them and based on the example of different drivers with different driving styles. However, the paper never mentioned a multi-expert scenario and also doesn't try to cluster the demonstrations but only randomly removes few demonstrations; thus, it is not even clear how the trajectory classes would tackle such problem.
4) Bad structure. While it is in general a legitimate approach to introduce the algorithm step by step, I think you should always also provide a rough sketch of the overall algorithm already early on, so that the reader has some context when you introduce the details. The way of introducing the trajectory classes is also a good example for the bad structure of the paper. Before introducing them, there is no mentioning of them, there is no motivation for them, and for all I can tell the paper doesn't even discuss the problem that they are supposed to tackle. And then in Section 2, they are introduced by stating "Suppose that we have trajectory classes [paraphrased]" and moving on as if nothing happened.
5) I must say that the writing is also quite bad. Some sentence are hard to follow due to grammar mistakes, especially if the reader doesn't have enough background to infer their meaning. For example "[...] suffer from the problem that the true reward shaped the changing environment." should probably mean something like "suffer from the problem that the learned reward function is shaped---that is, it is entangled with the dynamics---and, thus, does not transfer to different environments". As another example "However, GAIL is in a lack of an explanation of expert’s behavior and a portable representation for the knowledge transfer which are the merits of the class of the MaxEnt approach, because the MaxEnt approach is equipped with the "transferable" regular structures over reward functions."

__Soundness__
It doesn't make sense to talk a lot about the soundness of the approach before clarifying what it actually does. However, it seems to me that the learned distribution is only heuristic and does not approximate the posterior for any given prior. Also, for the linear reward function, the maximum likelihood objective is convex, so I think that the different weights should even converge to the same solution.

__Evaluation__
I don't demand MuJoCo experiments and the like, and in some cases evaluations on gridworlds can be sufficient. But I must say, that the excuse for not performing continuous control problems because the reward functions can not be visualized by 2d-heatmaps lured out a smile when reading the article. If I understand the algorithm correctly, it is significantly more expensive than standard MaxEnt-IRL which already doesn't scale to such problem settings as it requires iteratively solving the reinforcement learning problem.
The results on the objectworld are also not convincing. The expected value difference is for 80 demonstrations larger than for 40, and for 160 demonstrations larger than for 80. Similary, the performance initially degrades when increasing trajectory lengths and only improves again when taking trajectories of length 32 or more.
The sentence "A notable point in Figure 3 is that very few expert demonstrations (less than 200) for our approach also yields a small EVDs, which manifests the merit of Monte Carlo mechanism in our approach." is highly misleading. 200 trajectiories are not "few" for a 10x10 gridworld and an EVD over 20 is not small. When introducing this environment, Levine et al. (2011) achieved an EVD of around 1 based on 8 demonstrations (using 8 steps instead of 5).
The robustness experiment seems to subsample the weights based on the EVD (which requires knowledge of the true reward) before evaluation.

Questions
--------------
To better understand the proposed algorithm, I have the following questions.
1) How does the algorithm relate to MCEM? Is it some sort of EM within MCEM, where the first stage corresponds to the Monte-Carlo E-Step and the EM in the second stage corresponds to M-step of MCEM?
2) How are the densities g and h defined?
3) How does $\Omega_1$ affect the optimziation. How does it differ from $\mathcal{W}$?
4) How exactly are the m-step update steps performed? Do they indeed correspond to m gradient steps on the MaxEnt objective?
5) Section 3.3.: Should the inequality hold for $\lim_{t \to \infty}$?
6) What's the difference between DSIRL and SIRL?
7) How do the algorithm compare in terms of computational cost?

Assessment
----------
Unfortunately, I don't see a chance for acceptance here. Even if the algorithm was sound and sufficiently novel, the paper would need to be almost completely rewritten in order to address severe problems of the current presentation.

---

### Official Review · AnonReviewer3 · 2020-10-28
**Review for Paper78**

**Rating:** 2
**Confidence:** 5

**Review:**

### Summary

The authors proposed inverse reinforcement learning (IRL) algorithm based on Monte Carlo expectation-maximization (MCEM) that maximizes the predictive distribution of trajectories given the reward distribution parameter (eq (1)). In my understanding, the knowledge of the environment dynamics is assumed. The authors tried to validate the proposed idea on objectworld (Levine et al., 2011)

### Quality
The quality needs to be improved in the sense that a clear theoretical link between the target problem and MC-EM cannot be found in the submission. For example, the main objective (3) is optimized through (4) and (5), but the relation is unclear. There are lots of such things in the submission.

### Clarity
The readability of the submission is poor and needs to be improved. Lots of terms are unclear to me (e.g., succinctness, robustness, transferability of rewards). At some part of derivation, I couldn’t understand the motivation. Experiment settings are unclear, and the results are not confident and seem irreproducible with given information.

### Originality
Exploiting the distribution of reward is considered in Bayesian IRL. I think the probabilistic view was originated from Bayesian IRL (e.g., uniform prior on rewards may cover the idea of this work). The submission only sets MaxEntIRL as its baseline, but I think Bayes IRL should have been considered.

### Significance
There seems to be a minor contribution

### Detailed comments
(p.1, `Abstract`) `expert demonstrations may be optimal for many policies`
- I feel this statement is weird since we haven’t defined the optimality of expert demonstrations.

(p.1, `Abstract`) `we generalize the IRL problem to a well-posed expectation optimization problem stochastic inverse reinforcement learning (SIRL) to recover the probability distribution over reward functions.`
- SIRL tries to solve the inherent issue of IRL problem, not **generalize** IRL. Also, since Bayesian IRL also recovers the reward distribution, I couldn’t get the major advantage of the SIRL from this statement.

(p.1, `Abstract`) `The solution is succinct, robust, and transferable`
- Definitions of these expressions seem ambiguous to me.

(p.1, `Abstract`) `a global viewpoint`
- Again, ambiguous.

(p.1, `Introduction`)
- It would be better to write it in a more abstract way and separately write down the `Related Work` section.
- References should be much clearer: LaTeX commands like `\citet{}` and `\citep{}` should both be used.

(p.1, `Introduction`) `if the model dynamics are known`
- Recent works on IRL such as adversarial IRL (Fu et al, 2017) didn’t require the knowledge of model dynamics.

(p.1, `Introduction`) `The recovered reward function provides a succinct, robust, and transferable definition of the learning task`
- `succinct, robust, and transferable`: Ambiguous

(p.1, `Introduction`) First paragraph
- Lots of words from `Abstract` seem to be repeated.

(p.1, `Introduction`) `In a real-world scenario, experts always act sub-optimally or inconsistently, which is another challenge.`
- The sentence seems abrupt. The terms like `sub-optimal` and `inconsistent` here are awkward.

(p.1, `Introduction`) `imposes regular structures of reward functions in a combination of hand-selected features`
- GAIL (Ho et al, 2016) doesn’t require a hand-crafted feature.

(p.1, `Introduction`) `hand-selected by experts`
- A word `experts` here seems to imply a reward designer, not an expert on target tasks. I’d rather use a different word here.

(p.1, `Introduction`) `based on demonstrations respectively`
- `respectively` seems inappropriate.

(p.1, `Introduction`) `Influenced by the work of Finn et al. (2016a;b)`
- How these references affected AIRL needs to be mentioned.

(p.2, `Introduction`) `because the MaxEnt approach is equipped with the "transferable" regular structures over reward functions.`
- In Ziebart et al., 2008, transferability wasn’t mentioned.
- I believe the statement -- MaxEnt itself gives transferable reward feature -- is wrong but you should share the correct reference if this is true.

(p.2, `Introduction`) `The solution of SIRL is succinct and robust for the learning task in the meaning that it can generate more than one weight over feature basis functions which compose alternative solutions to the IRL problem`
- This explanation seems insufficient to understand the meanings of “succinctness” and “robustness”.

(p.2, `Introduction`) `Benefits of the class of the MaxEnt method,`
- Thanks to the benefits of the class of the MaxEnt method?

(p.2, `Introduction`) `Since of the intractable integration in our formulation,`
- Due to the intractable integral in our formulation?
- I think the intractability of the mathematical derivation didn’t need to be mentioned in `Introduction`.

(p.2, `Introduction`) `in a model-based environment`
- when model dynamics is known?

(p.2, `Introduction`) `In general, the solutions to the IRL problem are not always best-fitting in the previous approaches because a highly nonlinear inverse problem with the limited information is very likely to get trapped in a secondary maximum in the recovery.`
- I couldn’t understand what the authors wanted to emphasize.
- It seems like they intended to emphasize the problem of local optima, but I don’t know if such a problem is exactly what’s happening in IRL.

(p.2, `Introduction`) `global exhaustive search`
- What does `global` imply? Knowledge of dynamics?

(p.2, `Introduction`) `theoretically convergent demonstrated by pieces of literature`
- is theoretically convergent?
- How the theorem in the references (Caffo et al., 2005, Chan and Ledolter, 1995)  is applicable to the proposed idea should be much clearer since this is one main advantage that the authors argue. For example, what kind of assumptions are required to acquire global optimality? What is the algorithmic assumption of MC-EM for optimality? How are those assumptions linked with IRL setting?

(p.2, `Introduction`) `is also quickly convergent`
- converges quickly?
- How can we guarantee the convergence speed? Empirically or theoretically?

(p.2, `Introduction`) `the preset simple geometric configuration over weight space in which we approximate it with a Gaussian Mixture Model (GMM)`
- preset -> predefined?
- approximate it -> approximate

(p.2, `Introduction`) `We generalize the IRL problem`
- It seems the objective is not a generalization.

(p.2, `Preliminary`) $\mathcal{T}:=\mathbb{P}(s_{t+1}=s’|s_t=s, a_t=a)$
- $\mathcal{T}(s’|s, a):=\mathbb{P}(s_{t+1}=s’|s_t=s, a_t=a)$

(p.2, `Preliminary`)  `a sequential of state-action pairs`
- a sequence of state-action pairs?

(p.2, `Preliminary`) `The estimated complete MDP yields an optimal policy that acts as closely as the expert demonstrations.`
- The discount factor should be considered as well.

(p.3, `Regular Structure of Reward Functions`) $\mathcal{N}$
- I’d rather use a different letter since $\mathcal{N}$ is used to indicate Gaussian distribution in Section `Second Stage`.

(p.3, `Regular Structure of Reward Functions`) $\{\phi_i(s, a)\}_{i=1}$
- $\{\phi_i(s, a)\}_{i=1}^M$?

(p.3, `Problem Statement`) $\mathrm{MDP}\backslash R:=(\mathcal{S}, \mathcal{A}, \mathcal{T}, \gamma)$
- The definition doesn’t match with one without the discount factor $\gamma$ in `Preliminary`.

(p.3, `Problem Statement`) $\{\phi_i(s)\}_{i=1}^M$
- $\{\phi_i(s, a)\}_{i=1}^M$?

(p.3, `Problem Statement`) weights $\mathcal{W}$
- The definition should be provided.
- Either $\mathcal{W}=(\alpha_1, …, \alpha_M)$ (for linear model) or the weights of neural network (for non-linear model)?

(p.3, `Problem Statement`) `more likely generates weights to compose reward functions as the ones derived from expert demonstrations`
- Is this only a special case of Bayesian IRL?

(p.3, `Problem Statement`) `Suppose a representative trajectory class ~`
- The explanation should be clarified. In my understanding, $\mathcal{C}_\epsilon^E$ is a class of sets of trajectories.
- Why do we need to care such a class with $\epslion$ threshold?

(p.3, `Problem Statement`) Integrate out unobserved weights $\mathcal{W}$
- What does *unobserved* weights mean?
- Integrate out -> Marginalizing out?

(p.3, `Problem Statement`) trajectory element set $\mathcal{O}$ assumes to be uniformly distributed for the sake of simplicity in this study
- I don’t fully understand what’s the advantage of considering a representative trajectory class and why it is required.
- The section `Note:` tries to explain it, but more explanation or theorems seems to be needed. How can we theoretically guarantee that using a representative trajectory class doesn’t affect our estimation? It seems to me that we cannot guarantee the optimality with this class is the same as the original optimality.

 (p.3, `Problem Statement`) $f_{\mathcal{M}}$
- How this quantity is related to reward weights is unclear to me. The relationship between weights and $f_\mathcal{M}$ for both linear and non-linear models should be specified.

(p.3, `Note:)
- Instead of using a separate section, I’d rather put these statements in the middle of `Problem Statement` for a clearer explanation.

(p.4, `Two-stage Hierarchical Method`)
- Why do we need to use two-stage method instead of single-stage method (joint optimization over $\Theta_1$ and $\Theta_2$)? The advantage of two-stage methods should be briefly mentioned when it first appears for readability.
- How does the iterative update rule (4), (5) guarantee the optimization of (3)? It’s unclear to me due to the expectation in (4) and (5). My guess is that direct optimization of RHS of (3) is not possible and (4) and (5) might be either lower or upper bound of (3) due to Jensen’s inequality.

(p.4, `Initialization`) `~in each learning task`
- Do we care about multi-task learning or multiple reward weights only? I believe the latter case.

(p.6, `Experiments`) `since almost only objectworld provides a tool that allows analysis and display the evolution procedure of the SIRL problem in a 2D heat map, we skip the typical invisible physics-based control tasks for the evaluation of our approach, i.e. cartpole Barto et al. (1983), mountain car Moore (1990), MuJoCo Todorov et al. (2012), and etc.`
- I think this makes the contribution weaker. At least a few classic control tasks should have been considered. One way of evaluating the quality of rewards is retraining the agent with acquired reward, which is already widely used in the literature.

(p.6, `Objectworld`)
- One figure for illustration will enhance readability.

(p.7, `Evaluation Procedure and Analysis`) `DSIRL`
- DSIRL abbreviates Deep SIRL but wasn’t mentioned.

(p.7, `Recovery Experiments`)
- How many runs were used? How’s the mean and confidence interval of the empirical result?

(p.8, `Robustness Experiments`)
- I couldn’t understand how the robustness of reward is related to the proposed experiments. How the robustness is defined and its relation to the experiment should be clarified.

(p.8, `Hyperparameter Experiments`)
- How is the range of hyperparameter search for all methods? Currently, only the results for SIRL and DSIRL are given.

(p.8, `Conclusion`)
- It seems like both succinctness and transferability were not discussed in the main part of the submission.

### References
- Levine et al., 2011, “Nonlinear inverse reinforcement learning with gaussian processes“
- Fu et al., 2017, “Learning robust rewards with adversarial inverse reinforcement learning”
- Ho et al., 2016, “Generative adversarial imitation learning”
- Ziebart et al., 2008, “Maximum Entropy Inverse Reinforcement Learning”
- Caffo et al., 2005, “Ascent-based Monte Carlo expectation-maximization”
- Chan and Ledolter, 1995, “Monte Carlo em estimation for time series models involving counts“

---

### Official Review · AnonReviewer2 · 2020-10-28

**Rating:** 4
**Confidence:** 2

**Review:**

The authors propose an approach to model-based inverse reinforcement learning which estimates a Gaussian mixture model over reward-function parameters. The method uses MCEM and samples reward functions from a current estimate of the GMM, updates them via a gradient-descent based maximum likelihood approach and then updates the GMM to fit the updated parameters. The authors evaluate the approach on objectworld.

* The paper is at times hard to follow and should be rewritten to be more clear. The contributions and assumptions could be stated more clearly and the paper would strongly benefit from proof-reading. It would also be helpful to disentangle the machinery of MCEM from the novel algorithmic contributions of this paper.
* The choice of a GMM to represent the distribution of parameters is not motivated at all in the paper. Intuitively, the main benefit is to allow for k reward-function archetypes that represent the set of expert trajectories well; however, there are no examples nor any evaluation to show in which case this is beneficial.
* DSIRL is used as an acronym for a variant of the method but is not defined in the paper as far as I can tell.
* While the method can seemingly be applied to deep as well as linear representations, it is unclear what the chosen features and representation is in the experiments.
* The method appears to draw a set of weights W_i as well as a corresponding set of expert trajectories O_i at random in each iteration. The motivation for the use of O_i is that it may correspond to different modes in the expert set, e.g. demonstrations by different experts; however, the assignment of weights to object sets is not consistent between iterations so it is unclear to me how this would be able to handle different experts.
* In the beginning it is mentioned that IRL methods require knowledge of the transition model. While many methods do, modern IRL methods are model-free more often than not, so this claim is misleading.

---

### Official Review · AnonReviewer4 · 2020-11-02
**Less theoretical innovation and toy experiments only**

**Rating:** 3
**Confidence:** 3

**Review:**

The authors propose to solve underspecified IRL problem  using an MCEM approach. They claim it is the first succint, robust and transferrable solution and have some results on a Gridworld-like environment. Assessment: The MCEM framework actually is a good fit to the IRL problem and I dont recall if this has been explicitly called out in the literature before. But to me, the connection and resulting algorithm don't seem enough it terms of innovation and usefulness. Arguably, both BIRL and MaxentIRL do essentially (sampled) EM with different parametric forms. The experiments are unacceptably small scale and inconclusive.

There were confusing parts in the exposition that I will outline below.

Going by section:

Intro: A good overview of the IRL literature.

sec 2: In the problem statement itself you need to say something about how the expert demonstrations are connected to R. Otherwise they seem like there is no relation at all.

2.2:
I'm trying to understand the motivation behind the definition of C^E_\epsilon.

In the objective formulation you are setting \Theta to maximize the likelihood of a randomly choose subset of expert trajectories. I have never seen such a thing done before and it is interesting and possibly a route to robustness, but I would have liked more explanation and discussion. In particular, the choice of a uniform measure across subsets of different cardinalities seems bad.

 Would it be simpler to just introduce a hidden variable for each trajectory, stating whether it is valid or not (i.e.  drawn from the true reward function and should therefore be used in likelihood computation). I agree then with the 3rd point in 2.2.1, this could be a different way to model expert sub-optimality than BIRL.

3.1.2:
    It took me a while to read this and I dont think I understand. You seem to set \Theta_2 = W by the end. Then this seems to be very complicated and redundant. In particular, it is very confusing how the dimensionality of the parameter space can depend on the number of monte carlo samples and changes at each iteration ! Maybe I misunderstood something.

3.1.3:
  This seems like a straightforward EM derivation.

3.2:

 What is the difference between the 2 stopping criteria? Just the fact that the 2nd has a patience of 3? if so, seems trivial.

3.3:

 what is being formally claimed here? That given the condition on N_t, convergence will hold and proof is in one of the references?

4 (Experiments):

  Experiments are done on exactly 2 small instances of a variant of Gridworld. This is much smaller scale than the experiments you would expect to see in an ICLR paper (even granted that ICLR is very theory-focused). Previous papers have used IRL for real world problems like robotics, vehicle routing etc.

Summation: A couple interesting starting point ideas in the theory part, but not completely fleshed out and ultimately the development of the EM approach is partly straightforward and part of it is very confusing to me (sec 3.1.2).  Experiments are very unconvincing in terms of scope.

---

### Official Review · AnonReviewer5 · 2020-11-04
**Promising idea, but clarity lacking and evaluation weak**

**Rating:** 3
**Confidence:** 4

**Review:**

The paper proposes a novel method for inverse reinforcement learning: inferring a (distribution over) reward functions from a set of expert demonstrations. Prior work has either learned a point-estimate, notably maximum entropy IRL, or used Bayesian methods to learn a probability distribution over reward functions. Maximum entropy IRL has scaled to complex environments with unknown dynamics and non-linear rewards (with methods such as AIRL), but do not learn a probability distribution. By contrast, Bayesian IRL is more theoretically principled, but has not scaled to complex environments or non-linear rewards. This paper performs maximum likelihood estimation of a parameter for a *generative model* over probability distributions, using a Monte-Carlo expectation-maximization (MCEM) method. It therefore still outputs a probability distribution like Bayesian IRL, but is able to learn non-linear rewards unlike prior Bayesian methods.

Strengths:
  - The method is novel.
  - The motivation of the work is good: you identify important problems on IRL in the first introductory paragraph. (It would benefit from more follow-through -- which of these problems have you solved?)
  - Reporting examples of learned reward functions (or mean of the distribution learned) as well as summary statistics like EVD gives a more thorough understanding of the method.

Weaknesses:
  - The paper was difficult to understand.
  - The contribution seems limited. The method still seems challenging to scale (and no attempt is made to evaluate this), which seems to be it's main advantage relative to Bayesian IRL. Moreover, it is unclear if the probability distribution learned is well-calibrated, unlike Bayesian IRL.
  - Weak baselines. DeepMaxEnt is a fairly old approach -- why not try e.g. AIRL? There's also no information as to how you trained the baselines, so it's difficult to know whether e.g. hyperparameter tuning was performed appropriately (noting that this environment is fairly different to what DeepMaxEnt was originally trained in). You should report the results of GPIRL given that this algorithm was developed for exactly this environment.

I find the approach intriguing and would encourage you to continue developing it, but the submission is too preliminary to accept at this stage. In particular, I would suggest the following modifications:
  + Significant edits to improve clarity. For example, the abstract is quite terse, especially the second sentence. I understood it since I am very familiar with this area, but most readers would be lost. I had to read section 2.2 several times to understand what you were proposing -- here (and elsewhere) you would benefit from giving the reader some intuition before diving into the math. The basic idea is relatively simple: you are performing maximum likelihood estimation on parameters $\Theta$ that define a probability distribution over weights. Then there are some details: how you sample the data when you perform MLE (effectively from a power set of the demonstrations, thresholded by some minimum size),
  + Clarify what the benefits of your method relative to prior work are, and then rigorously justify this (whether theoretically or empirically). For example, is having a probability distribution over rewards actually a desiderata (in which case you should evaluate if they're well-calibrated), or is it an artifact of the method? Likewise, being candid about it's limitations would help the reader evaluate whether it is appropriate for their application or what novel research directions exist to improve it.
  + Improved evaluation. Stronger baselines as discussed in "Weakness" above. More environments. Perhaps report runtimes -- this would help assess scalability. I found this line uncompelling: "Note that since almost only objectworld provides a tool that allows analysis and display the evolution procedure of the SIRL problem in a 2D heat map, we skip the typical invisible physics-based control tasks for the evaluation of our approach". First, this is wrong: objectworld is not that unique, you could visualize reward over e.g. a gridworld or tabular MDP, and many IRL papers have done so. Second, providing this visualization does not preclude also evaluating in more complex environments -- as the AIRL paper did, for example. I strongly suspect your algorithm simply won't scale to such environments (at least without considerable hand-designed feature engineering) -- if this is true then admit it and discuss how you can address it in future work, and if it's false then show that I'm wrong with results.
  + You should discuss limitations of your work somewhere, e.g. conclusion. For example the convergence guarantee only holds in the limit of infinite data if I understand -- there is no finite-time bound? This is common for MCMC methods which your approach is related to, so not too surprising, but it's important to make the reader aware of.

Some questions I would appreciate clarification on:
  - How is $f_M$ actually defined? In particular, what do the weights $\mathcal{W}$ really do? A naive reading would suggest that you take a linear combination of the "feature basis functions" -- but I think you must do something more complex since you evaluate in an environment with a highly non-linear reward?
  - What is Algorithm 1: Generative Algorithm actually meant to be doing? It seems one could not use this in practice, since it requires computing the EVD, which is only computable if one knows the ground-truth reward -- in which case no need for IRL. So I assume it is meant solely for evaluation -- but I am unsure what this is evaluation. If this is meant to be focusing on the most robust samples, then you should report what probability mass these samples have. Also, is "any W'" meant to be a universal or existential quantifier?

Some points to improve clarity. At a high-level:
  1. Paper might benefit from separate related work section. Intro is currently serving as this. But it detracts from the story. You could just summarize the IRL problem, what the key deficiency in existing work is that you're trying to solve, and then dive into describing your method and contributions.
  2. Many parentheses should be parenthetical, e.g. “are known Russel (1998)” -> “are known (Russel, 1998)” – change \cite to \citep in the source.
  3. As mentioned before, would benefit from proofreading for grammar. Examples: “problem is ill-posed that the” -> “problem is ill posed: the”, “an inverse problem that a” -> “an inverse problem where a”; “is in a lack of an explanation” -> “does not explain”; “variability set” -> “variable set”.

Some specific points:
  - Abstract, “considerable performance” – weaselword, how good is it relative to expert, to a baseline, ...?
  - Preliminary, description of transition dynamics: “being current state s, taking action a and yielding next state s'” – does not make the conditionality clear. Perhaps “transitioning to next state s' when taking action a in state s”.
  - Preliminary, MDP\R: I have normally seen this include the discount factor, which is important to understand expert behavior. I think you assume this too in your algorithm? If not, flag this, and describe how you recover rewards without knowing the discount.
  - Typo: "experimental trail"->"experimental trial".

---

### Decision · Program_Chairs · 2021-01-07
**Final Decision**

**Decision:**

Reject

**Comment:**

This paper describes a method called 'stochastic' inverse reinforcement learning. It is somewhat unclear how this differs from other probabilistic approaches to IRL. In particular Bayesian approaches have been used in the past to obtain distributions over reward functions. However, SIRL tries to estimate a generative model over such distributions. All the reviewers foudn the paper suffering from lack of clarity, in particular with respect to how the model/algorithm is constructed. There are some possible technical problems with respect to claims about inferring demonstrations by different experts (cf. work on multi-task IRL). The experiments also seem to be insufficient.